# Intention to use vasectomy and its associated factors among married men in Debre Tabor Town, North West Ethiopia, 2019

**Alemu Degu Ayele** [1], **Fentahun Yenealem Beyene**[2] *, **Kihinetu Gelaye Wudineh**[2], **Bekalu Getnet Kassa**[1], **Yitayal Ayalew Goshu**[1], **Gedefaye Nibret Mihretie**[1]

**1** Department of Midwifery, College of Medicine and Health Sciences, Debre Tabor University, Debre Tabor, Ethiopia, **2** Department of Midwifery, College of Medicine and Health Sciences, Bahir Dar University, Bahir Dar, Ethiopia

* yenefenta84@gmail.com

**Data Availability Statement:** All relevant data are within the manuscript and its Supporting Information files.

## Abstract

### Background

Vasectomy is one of the most effective and permanent male contraceptive methods, and involves cutting and ligating the vas deferens to make the semen free of sperm during ejaculation. Although it is effective, simple, and safe, it is not well known and practiced in the majority of our community. This study assessed the intention to use vasectomy and its associated factors among married men in Debre Tabor Town, North West Ethiopia, 2019.

### Methods

A community- based cross-sectional study was conducted among 402 married men from March 05 to April 15, 2019. A simple random sampling technique was employed to select the study participants. Data was collected by face to face interview using a structured and pre-tested questionnaire. Questions concerned socio-demographic and reproductive variables and views on vasectomy. The association between variables was analyzed using a bivariable and multivariable logistic regression model.

### Result

A total of 402 participants were included with a response rate of 98.75%. The mean participant age was 37.12(SD ± 6.553) years with the age range of 20–56 years. The prevalence of intention to use vasectomy was 19.6% with 95%CI (15.6%-23.4%). Multivariable logistic regression showed that age from 30–39 years (AOR = 3.2(95% CI: 1.19–8.86)), having more than three living children (AOR = 2.5(95% CI: 1.41–4.68)), good knowledge (AOR = 3.4(95%CI: 1.88–6.40)) and positive attitude (AOR = 4.8(95% CI: 2.61–8.80)) of married men were significantly associated with intention to use vasectomy.

### Conclusion and recommendation

Intention to use vasectomy was comparable with findings in four regions of Ethiopia (Amhara, Oromia, SNNP, and Tigray). Age, the number of living children, knowledge, and

**Funding:** Yes, Bahirdar University.

**Competing interests:** The authors have declared that no competing interests exist

attitude were significantly associated with the intention to use vasectomy. Improving the level of knowledge and attitude towards vasectomy is an essential strategy to scale up the intention of men to use vasectomy.

## Background

In mid-2016, the estimated population of Ethiopia was 101.7 million which is the second fast population growth in Africa next to Nigeria [1]. It is also affected by high maternal morbidity and mortality related to unintended and unwanted pregnancies [1, 2]. The goal of family planning all over the world has attracted attention due to its importance in decision making about population growth and development issues [3]. Worldwide, using contraceptives potentially reduced maternal mortality by 44% [4].

Even though the 1994 International Conference on Population and Development (ICPD) in Cairo emphasized that men's involvement in sexual and reproductive health issues is very important for a better outcome and set clear directions to increase men's participation in family planning, male involvement in family planning is still very low in Africa [5–7].

Vasectomy is a male method of contraceptive which is known and acceptable only in some developed countries of the world. Although it is safe and easy to perform, only 45 million couples worldwide rely on it [8, 9].

International health organizations in recent years have suggested that the involvement of men on the utilization and promotion of contraceptive methods is very crucial to provide couples with more male-oriented contraceptive choices, such as vasectomy. Even though its procedure is simple and easy with a high success rate (>99%) and minimal complications, it is underutilized around the world, especially in developing countries including Ethiopia [10–12]

Couples who complete their family or no want more children, vasectomy is the option than tubal ligation, due to it poses fewer surgical risks. Although it's a higher effective, simple process and fewer complications, currently vasectomy users in the United States only 8% [13].

Ethiopia has a target plan to increase the contraceptive prevalence to 55% by the year 2020 [14]. To achieve this goal, the government in collaboration with other stack holders focuses on different FP methods by increasing demand and access to long-acting and permanent methods [15]. Due to this effort, the Contraceptive Prevalence Rate (CPR) with currently married women were reached 36%, with 35% using modern methods and 1% using traditional methods. Of 35%, injectable (23%), implant (8%), IUD(2%), pills(2%), male condom and tubal ligation <1% Ethiopia Demographic Health Survey (EDHS 2016) [16].

Different reviewed literature reports showed that the intention to use vasectomy was associated with different variables [11, 15, 17, 18]. Age, educational status, occupation, religion, Cultural beliefs, societal norms, lack of knowledge about the procedure for a vasectomy, and misconceptions were found to be predictors of intention to use vasectomy. In addition, the duration of married time, number of living children, complete family size, the future desire of more children, accessibility of service, level of knowledge, and attitude of men towards vasectomy have an influence on intention to use vasectomy [18–22].

According to EDHS 2016, the national demand for family planning is 58%. However, vasectomy utilization is almost negligible [16]. There are very few studies on the intention of married men to use vasectomy in Ethiopia. Therefore, this study was designed to determine the intention of men to use vasectomy and identify factors associated with the intention to use vasectomy among married men in Debre Tabor Town, Northwest Ethiopia.

## Methods

### Study design and setting

A community- based cross-sectional study design was conducted in Debre Tabor Town from March 05 to April 15, 2019. The Town is found in the Amhara region, and it is a capital city of South Gondar Zone, North West Ethiopia. It is located 665 kilometers from the capital city of Ethiopia and 103 kilometers North West of Bahir Dar Town. The Town was divided into six small administrative units called kebeles with a total population of 92, 530 based on the 2018 report of Town administration [23]. The Town had one general hospital, three health centers, and four health posts that provide family planning and other health services in the study area. All married men whose wives were in the reproductive age group living in all six kebeles of Debre Tabor Town were the study population. All married men whose wives were in the reproductive age group in the town within the study period were included. Whereas married men whose wife were in the reproductive age group who were critically ill (bed reddened), had already done vasectomy, had infertile wife, and had a wife with a hysterectomy and married men women who live less than 6 months during data collection period in Debre Tabor town were excluded.

### Sample size calculation and sampling procedures

Epi-Info version 7 statistical software was used to calculate the sample size assuming that 39% of men had the intention to use vasectomy [21], 95% confidence interval with a 5% margin of error. By adding a 10% non-response rate the final sample size was 402. A simple random sampling technique was applied to select 402 married men. A total of 14,088 households and 14,614 married men were living in the town [23]. A household was sampling unit in each kebeles and samples were allocated proportionally to each kebeles based on their total household number. Household numbers with married men in each kebeles were found from the kebeles registration book. Study households were selected from each kebeles through a simple random sampling technique by using a table of random numbers starting from kebele one from a random start point. The first household was selected in each kebele by using the lottery method. One married man per household was interviewed. When two or more eligible men were found in one household, only one was interviewed by lottery method and if no eligible men were identified in the selected household, the next eligible household located in the clockwise direction was visited and included until we got the desired sample size (Fig 1).

### Measurement and data collection procedure

Data were collected via face to face interview techniques using a structured, validated, and pretested questionnaire. The tool was first prepared in English then translated to Amharic and back to English by language experts to maintain the consistency of the instrument. Three diploma holder male midwives conducted the face to face interview and one BSc degree midwife supervised the data collection process. The information was collected on participants socio-demographic characteristics (age, religion, educational status, wife education, occupation and wife occupation), Reproductive characteristics (duration of married life, number of living children, the desire of more children, discussion with a partner, support from a partner), knowledge- related, and attitude- related questions (S1 File). Intention to use vasectomy means the planning of respondents to use vasectomy as a contraceptive method for a future time [24]. In this study knowledge about vasectomy was determined by using nine knowledge's related questions. A value of one and zero was given for each correct and incorrect answer respectively and labeled as good knowledge; those individuals who answered at least

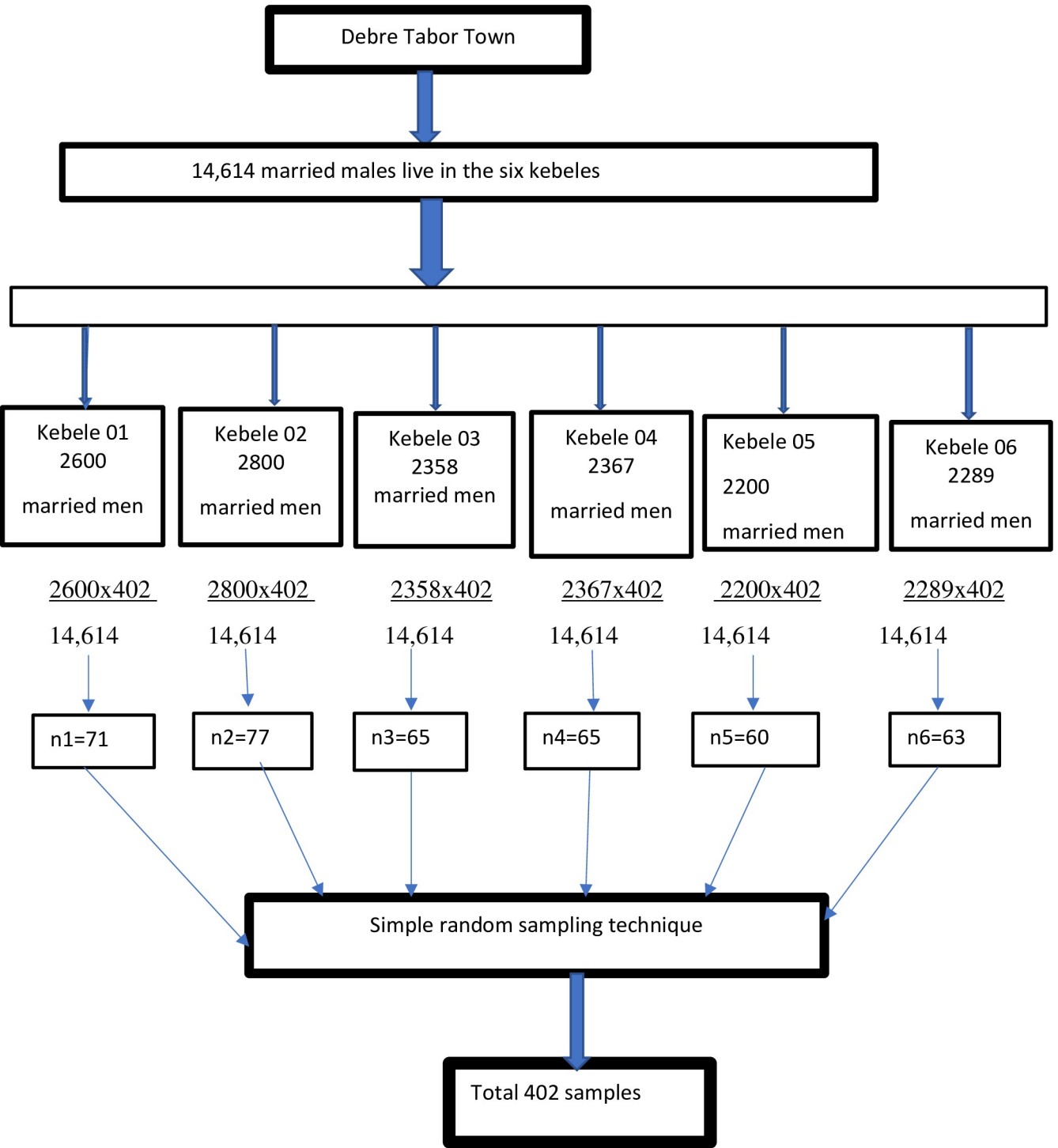

**Fig 1. Schematic presentation of the sampling procedure for a study conducted on an intention to use vasectomy and its associated factors among married men in Debre Tabor Town North West Ethiopia 2019.**

five knowledge related questions and poor knowledge; those answered less than five knowledge's related questions. In this study attitude about vasectomy was determined by using nine attitudes related questions and labeled as a positive attitude; those participants who scored

greater than or equals to the mean score and negative attitude; those individuals who scored less than the mean score [25, 26].

## Data quality assurance

The data collection tool was prepared after an intensive review of relevant literature. Pretest of the questionnaire was done on 20 married men in Wereta Town and adjustments were made accordingly. Any error, ambiguity or incompleteness identified was corrected immediately. Data collectors and supervisors were trained for one day about the contents of the questionnaire, the aim of the study, method of data collection, confidentiality, responders' right, and informed consent. The completeness of the data was checked by data collectors during data collection and also immediately after data collection by the supervisor and principal investigator.

## Data processing, analysis, and interpretation

Data were cleaned, coded, and entered by using Epi-Data version 3.1 and then exported to SPSS version 23 for analysis. Descriptive statistics like frequencies and percentages were used to present the categorical independent variables and mean/standard deviation was used to describe a continuous variable. Frequency tables and graphs were used to present descriptive results. Bivariable logistic regression analysis was executed by computing odds ratio (OR) with a 95% confidence interval to see the association between each independent and dependent variable. Finally, all independent variables associated with dependent variables with $p<0.2$ were entered into multivariable logistic regression for further analysis by controlling confounding factors and significant association was identified based on $p<0.05$ and adjusted odds ratio (AOR) with 95% CI. Model adequacy was checked using the *Hosmer* and *Lemeshow* goodness of fit test (p-value = 0.41).

## Ethics approval and consent to participate

Ethical clearance was obtained from the institutional review board of Bahir Dar University. A support letter was written to the Debre Tabor town administration and consent was obtained. Participants of the study were informed about the purpose, objectives, and their right to participate or not participate in the study. Privacy and confidentiality of the study participant were ensured by keeping all information anonymous. Written informed consent was obtained from each participant.

## Result

### Socio-demographic characteristics of participants

A total of 402 married men participated in this study with a response rate of 98.75%. Half of the respondents 200(50.3%) were belonged to the age group of 30–39 years with the mean age and standard deviation of 37.12 (±6.553) years and with the age range of 20–56 years. The majority of the participants, 387(96.7%) were Amhara by ethnicity and orthodox Christian followers, 355(89.4%). More than half of the participants, 252(63.4%) were their educational status was college and above and one hundred eight two (45.8%) participants wives have attained college and above in their educational status. One hundred seventy-three (43.6%) participants and one hundred thirteen was a civil servant by occupation (Table 1).

**Table 1. Percentage distribution of the study population by socio-demographic characteristics; Debre Tabor Town, North West Ethiopia, March 05-April 15, 2019 (N = 397).**

| Variables | Frequency | Percent |
|---|---|---|
| **Age 20-29years** | 57 | 14.4 |
| 30–39 years | 200 | 50.3 |
| 40–49 years | 119 | 30.0 |
| ≥50 years | 21 | 5.3 |
| Mean age ± SD | | 37.12±6.553 |
| **Ethnicity Amhara** | 384 | 96.7 |
| Oromo | 7 | 1.8 |
| Others* | 6 | 1.5 |
| **Religion Orthodox** | 355 | 89.4 |
| Muslim | 33 | 8.3 |
| Others** | 9 | 2.3 |
| **Educational status No formal education** | 36 | 9.1 |
| Primary | 36 | 9.1 |
| Secondary | 73 | 18.4 |
| College and above | 252 | 63.4 |
| **Wife education No formal education** | 56 | 14.1 |
| Primary | 65 | 16.4 |
| Secondary | 94 | 23.7 |
| College and above | 182 | 45.8 |
| **Occupation Civil servant** | 173 | 43.6 |
| Private business | 164 | 41.3 |
| Employed at private | 45 | 11.3 |
| Daily laborer | 15 | 3.8 |
| **Wife occupation House wife** | 157 | 39.5 |
| Civil servant | 113 | 28.5 |
| Private business | 92 | 23.2 |
| Employed at private | 21 | 5.3 |
| Student | 14 | 3.5 |

*Tigray, Gurage
** protestant, catholic

## Reproductive characteristics of participants

Among the respondents, 157(39.5%) were living with their wives from 6–10 years with the average duration of marriage 9.85 years. Two hundred forty-three (61.2%) of participants had three living children. The majority of the study participants, 355(89.4%) were discussed about FP with their partner. One hundred two (25.7%) participants got emotional support from their partner to use vasectomy. Two hundred eighty-one (70.8%) participants were completed their family size. Among 116(29.2%) participants who had not completed their family size, 83(72%) participants were their number of children desire was three and less than three for their future life (Table 2).

## Knowledge of participants towards vasectomy

Two third of the study participants 247(62.2) had poor knowledge about vasectomywhile150 (37.8%) of them had good knowledge (Table 3).

**Table 2. Percentage distribution of the study population by reproductive characteristics; Debre Tabor Town, North West Ethiopia, March 05 -April 15, 2019(N = 397).**

| Variables | Frequency | Percent |
|---|---|---|
| Duration of married year ≤ 5 years | 95 | 23.9 |
| 6–10 years | 157 | 39.5 |
| 11–15 years | 77 | 19.5 |
| ≥16 years | 68 | 17.1 |
| mean ± (SD) | | 9.85±4.82 |
| Number of living children ≤3 children | 243 | 61.2 |
| >3 children | 154 | 38.8 |
| mean ±(SD) | | 3.01±1.56 |
| Discuss FP with a partner no | 355 | 89.4 |
| yes | 42 | 10.6 |
| Get emotional support from partner no | 102 | 25.7 |
| Yes | 295 | 74.3 |
| Complete family size no | 116 | 29.2 |
| yes | 281 | 70.8 |
| Future desire no of children(N = 116) | | |
| ≤ 3 children | 83 | 72 |
| >3 children | 33 | 28 |
| mean±(SD) | | 2.73±1.42 |

## The attitude of participants towards vasectomy

The majority of participants 245(61.7%) had a negative attitude while 152(38.3%) had a positive attitude towards vasectomy (Table 4).

**Table 3. Percentage distribution of the study population by knowledge related questions; Debretabor Town, North West Ethiopia, March 05-April 15, 2019 (N = 397).**

| Knowledge related questions | Response | Frequency (%) |
|---|---|---|
| Heard about vasectomy Do you know that vasectomy is a contraceptive method by cutting and ligating Vass deference? | Yes | 185(46.6) |
| | No | 212(53.4) |
| Vasectomy is permanent and irreversible? | Yes | 146(36.8) |
| | No | 251(63.2) |
| Does vasectomy require a minor surgical procedures? | Yes | 145(36.5) |
| | No | 252(63.5) |
| Seminal fluid during ejaculation are present after vasectomy | Yes | 88(22.2) |
| | No | 209(77.8) |
| Do you know how a vasectomy works? | Yes | 62(15.6) |
| | No | 335(84.4) |
| Vasectomy is done in Ethiopia without any charge | Yes | 28(19.6) |
| | No | 319(80.4) |
| Do you know where vasectomy service is available | Yes | 122(30.7) |
| | No | 275(69.3) |
| Have you heard that who can use vasectomy as a family planning? | Yes | 150(37.8) |
| | No | 247(62.2) |
| If yes who are they(N = 150) | All married men | 26(17.4) |
| | Married men who complete the family size | 124(82.6) |

**Table 4. Percentage distribution of the study population by attitude related questions; Debre Tabor Town, North West Ethiopia, March 05-April 15, 2019 (N = 397).**

| Attitude related questions | Agree | Neutral | Disagree |
|---|---|---|---|
| FP is the responsibility of women | 15(3.8) | 2(0.5) | 380(95.7) |
| Do you believe that vasectomy negatively affects sexual performance/desire | 95(23.9) | 222(55.9) | 80(20.2) |
| Do you believe that vasectomy has series side effects | 133(33.5) | 203(51.1) | 61(15.4) |
| Vasectomy is not acceptable in my religion | 280(70.5) | 52(13.1) | 65(16.4) |
| Vasectomy is culturally unacceptable | 259(65.2) | 63(15.9) | 75(18.9) |
| I am uncertain for the future pregnancy may happen after vasectomy | 82(20.7% | 204(51.3) | 111(28) |
| Vasectomy is similar to castration | 132(33.2) | 203(51.1) | 62(15.7) |
| Vasectomy can cause physical weakness, cannot do hard work | 38(9.6) | 247(62.2) | 112(28.2) |
| Vasectomy can ashamed the individual in the community | 82(20.6) | 127(32%) | 188(47.4) |

### Intention to use vasectomy and reason not to use vasectomy among study participants

This study identified that 19.6% with 95% CI (15.6–23.4) participants had the intention to use vasectomy as a contraceptive method for their future life.

In this study, 80.4% of the participants had no intention to use vasectomy as a contraceptive method for their future life due to many reasons. Among this, lack of awareness (74%) was the major reason followed by religious prohibition (70.5%) (Fig 2).

### Factors affecting intention to use vasectomy among study participants

In binary logistic regression; age, participant wife occupation, duration of the married year, number of living children, and level of knowledge and attitude of participants had an association with intention to use vasectomy. In multivariable binary logistic regression analysis, after adjusting other co-variables by using backward likelihood stepwise method; age, number of living children, level of knowledge, and attitude had a significant association with intention of married men to use vasectomy.

Participants whose age group between 30–39 years were 3.2 times more likely to have the intention to use vasectomy [AOR = 3.2, (95% CI: 1.19–8.86)] as compared to Participants whose age lies below 30 years. Married men who had more than 3 alive children were 2.5 times higher the odds of intention to use vasectomy [AOR = 2.5(95% CI: 1.41–4.68)] as compared to men who had less than or equals to three alive children.

Moreover, study participants who had good knowledge about vasectomy were 3.4 times the odds of intention to use vasectomy [AOR = 3.4(95% CI: 1.88–6.40)] as compared with study participants who had poor knowledge about vasectomy. Participants who had a positive attitude towards vasectomy were 4.8 times more likely to use vasectomy as compared with their counterparts [AOR = 4.8(95%CI: 2.61–8.80)] (Table 5)

### Discussion

Within the sphere of family planning, vasectomy is very often ignored, despite being one of the safest, simplest, and highly effective and least expensive contraceptive methods [5]. The descriptive report of this study revealed that around 37.8% and 38.3% of participants had good knowledge and positive attitude towards vasectomy respectively. This implied that there was poor knowledge and attitude towards vasectomy in the study area; to improve this; the country should design and implement the following strategies in its program;—Provision of Information and Services for Men in clinic setup, Outreach Male Motivators, and Peer Educators;

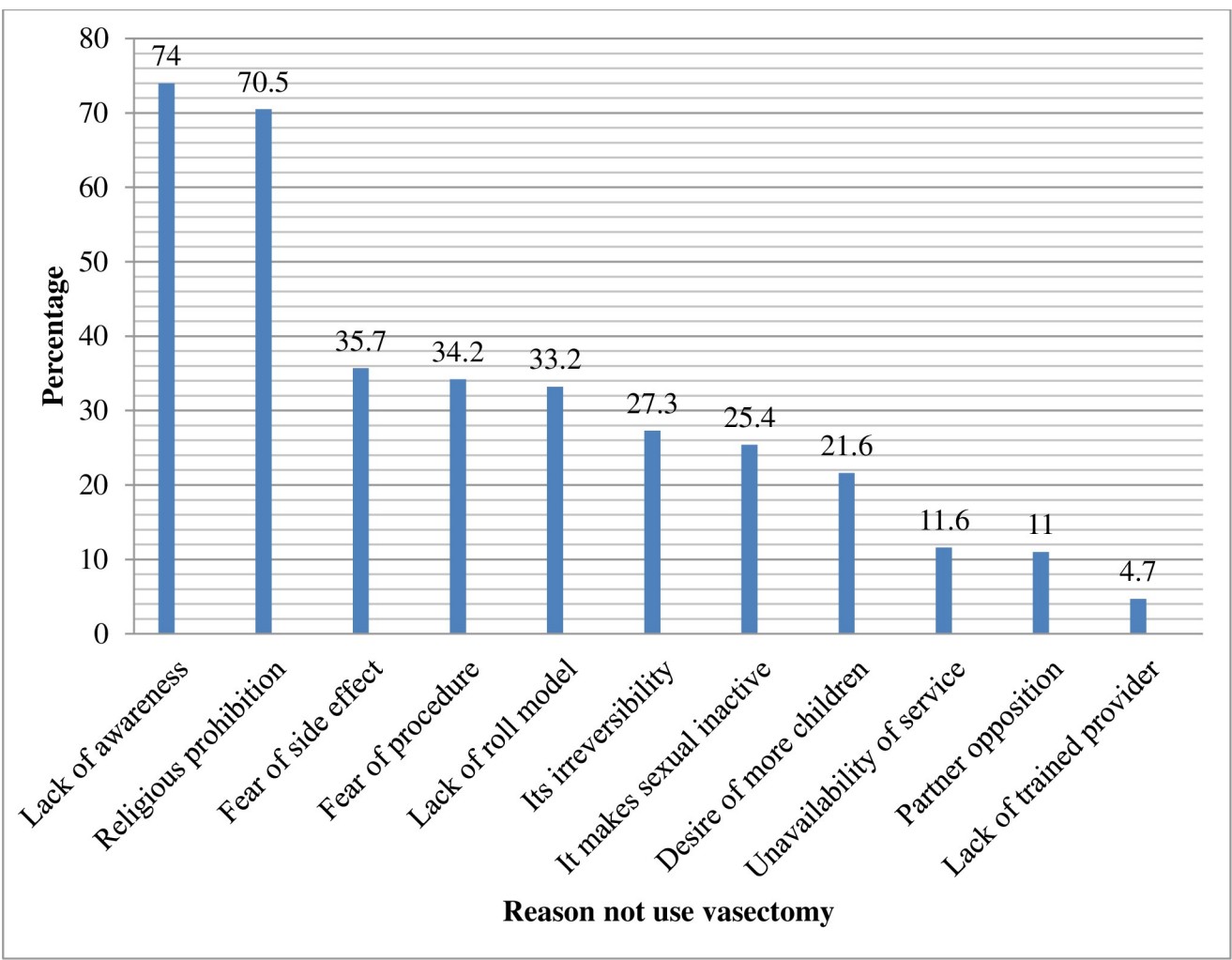

**Fig 2. Percentage distribution of study participants by reason didn't have the intention to use vasectomy for future life; Debre Tabor Town, North West Ethiopia, March 05-April 15 2019, (N = 319).**

Community Engagement; design community communication program; comprehensive Sexuality Education should be design and design community campaign. When compared to studies done in Malaysian private medical college which is (60.9%and 76%) of the respondents had a positive attitude and showed good acceptance towards vasectomy [27] which is significantly higher than from our studies. This might be due to the difference in study participants since the previous study was conducted on the medical students they might be hade a better view and attitude towards vasectomy than the general population.

Another surprising descriptive report of this study was participants educational level, college and above (63.4%) and participants occupation, civil servant (43.6%). This implies that as participants more educated and becoming civil servants; they might have easy access to information regarding to their fertility desire and might have more awareness and easily decide to use it. When compare to the finding in EDHS 2016 report which was (5%) and not more than 5% of respondents attained secondary school and above and their occupation is civil servant [16] respectively. This might be due to difference in study population, study area and time. The previous study was conducted at large country level which included both urban and

**Table 5. Bivariable and multivariable analysis of factors affecting intention to use vasectomy; Debre Tabor town, North West Ethiopia, March 05-April 15, 2019 (N = 397).**

| Variables | Intention to use vasectomy | | COR (95%CI) | AOR (95%CI) | P-value |
|---|---|---|---|---|---|
| | Yes | No | | | |
| Age 20–29 years | 6(10.5) | 51(89.5) | 1 | 1 | |
| 30–39 years | 55(27.5) | 145(72.5) | **3.224(1.309–7.938)** | **3.251(1.192–8.863)** | **0.021*** |
| 40–49 years | 12(10.1) | 107(89.9) | 0.953(0.339–2.684) | 0.861(0.274–2.702) | 0.797 |
| ≥50 years | 5(23.8) | 16(76.2) | 2.656(0.715–9.874) | 3.216(0.73714.028) | 0.120 |
| Educational status | | | | | |
| No formal education | 5(13.9) | 31(86.1) | 1 | 1 | |
| Primary | 6(16.7) | 30(83.3) | 1.24(0.342–4.498) | 0.99(0.237–4.13) | 0.989 |
| Secondary | 5(6.8) | 68(93.2) | 0.456(0.123–1.690) | 0.35(0.084–1.449) | 0.148 |
| College and above | 62(24.6) | 190(75.4) | 2.023(0.754–5.429) | 1.142(0.373–3.492) | 0.816 |
| Wife education | | | | | |
| No formal education | 9(16.1) | 47(83.9) | 1 | 1 | |
| Primary | 7(10.8) | 58(89.2) | 0.63(0.218–1.819) | 0.54(0.141–2.063) | 0.368 |
| Secondary | 16(17.0) | 78(83.0) | 1.071(0.438–2.617) | 0.996(0.279–3.553) | 0.995 |
| College and above | 46(25.3) | 136(74.7) | 1.766(0.804–3.883) | 1.149(0.299–4.412) | 0.84 |
| Occupation | | | | | |
| Civil servant | 33(19.1) | 140(80.9) | 0.354(0.118–1.063) | 0.218(0.042–1.128) | 0.069 |
| Private business | 31(18.9) | 133(81.1) | 0.350(0.116–1.055) | 0.257(0.049–1.337) | 0.106 |
| Employed in the private sector | 8(17.8) | 37(82.2) | 0.324(0.090–1.172) | 0.293(0.049–1.768) | 0.181 |
| Daily laborer | 6(40.0) | 9(60.0) | 1 | 1 | |
| Wife occupation | | | | | |
| House wife | 27(17.2) | 130(82.8) | 1 | 1 | |
| Civil servant | 28(24.8) | 85(75.2) | 1.586(0.875–2.876) | 1.067(0.488–2.332) | 0.870 |
| Private business | 12(13.0) | 80(87.0) | 0.722(0.346–1.506) | 0.499(0.208–1.199) | 0.120 |
| Employed at private | 5(23.8) | 16(76.2) | 1.505(0.508–4.459) | 1.643(0.425–6.349) | 0.472 |
| Student | 6(42.9) | 8(57.1) | **3.611(1.159–11.25)** | 0.858(0.180–4.082) | 0.847 |
| Duration of married | | | | | |
| ≤ 5 yrs | 12(12.6) | 83(87.4) | 1 | 1 | |
| 6–10 yrs | 31(19.7) | 126(80.3) | 1.702(0.827–3.502) | 1.007(0.422–2.405) | 0.987 |
| 11–15 yrs | 19(24.7) | 58(75.3) | **2.266(1.021–5.026)** | 1.161(0.371–3.635) | 0.798 |
| ≥ 16 yrs | 16(23.5) | 52(76.5) | 2.128(0.933–4.856) | 2.124(0.614–7.342) | 0.234 |
| No of living children ≤ 3 | 36(14.8) | 207(85.2) | 1 | 1 | |
| >3 | 42(27.3) | 112(72.7) | 2.156(1.306–3.559) | 2.575(1.416–4.684) | **0.002*** |
| Discus with partner on FP | | | | | |
| No | 5(11.9) | 1 | 1 | 1 | |
| Yes | 73(20.6) | 282(79.4) | 1.916(0.727–5.047) | 1.77(0.573–5.469) | 0.321 |
| knowledge poor | 25(10.1) | 222(89.9) | 1 | 1 | |
| good | 53(35.3) | 97(64.7) | **4.852(2.850–8.260)** | **3.473(1.884–6.404)** | **0.001*** |
| Attitude Negative | 24(9.8) | 221(90.2) | 1 | 1 | |
| Positive | 54(35.5) | 98(64.5) | **5.074(2.967–8.676)** | **4.801(2.617–8.807)** | **0.001*** |

* P-value < 0.05 considered as statistically significant

remote rural areas that might be results lower educational level and civil servants whereas the current study was conducted at single town the educational characteristics might be more educated and also as they more educated the more being civil servant also since they are nearby information and access to education. And the other hypothesis might be the previous study

participants marital status were both married and un married respondents were considered whereas current study only on married respondents considered; this also might be reason of the difference.

This study showed that 19.6% of participants had the intention to use vasectomy for their future life. This finding was in lined with the studies done in India showed that 21.4% and 21% of the participants had the intention to use vasectomy [28, 29] respectively.

The intention of married men to use vasectomy in this study was also in agreement with study conducted in Bangalore rural population (17%) [30], Indonesia (16.6%) [20] and four regions of Ethiopia (Oromia, Amhara, SNNP and Tigray) (18.1%) [15].

But the finding of this study was lower than studies conducted in Kenya and Rwanda (27.5%, 26.6%) [18, 19] respectively. The discrepancy may be due to differences in the number of children the participants had since only 38.8% of participants in this study had more than three children but in the comparable studies (76.6% and 63.3%) of the participant had more than three children respectively. In addition, only 38.3% of the participant in this study had a positive attitude but in Rwanda, 60% of the participant had a positive attitude towards vasectomy which had a direct relationship with their intention.

According to our study, the intention to use vasectomy as a FP method was also lower than study conducted in the East Wollega zone of the Oromia region (30%) [22]. The discrepancy may be due to difference in study setting and participants. The current study was community-based but the previous study was done in a health institution and the participants were men who were visiting health institutions together with their partner for FP service, maternity and child health unit, which has a positive impact on their intention by increasing their awareness about the method.

The finding of our study was higher than the study conducted in India (11%) and Turkey (11.4%) [17] and [10] respectively. The variation may be due to difference in the marital status of study participants, in the current study only married men participated but in the previous study, the participants were both married and unmarried men who had a significant difference in their intention since thinking about vasectomy without being married is not ideal and visible. And also the difference is might be difference in socio-demographic characteristics, study area, and participants.

The result of this study revealed that being in the age category of 30–39 years had an increased intention to use vasectomy as a contraceptive method as compared to the age category of 20–29 years. This finding is supported by studies conducted in India and Nepal [17, 31]. Another study conducted in Indonesia also showed that the age of the respondent which was found between 30–39 had a significant relationship with intention to vasectomy [32].

The possible explanation for this finding might be participants in this age category (30–39) might have easy access to information and educated from the younger once, as the educational status increase, the knowledge might also scale up in the same fashion which had a positive impact on intention. The other reason might be men in the age category of 30–39 may have steady jobs, a greater number of living children, and stable family life from younger once. Another explanation those mens, whose aged 40 and above years might be thought that their wives near to menopause and no more have children at more.

According to our study finding, having more than three living children was the other predictor variable as it stated that having more than three children increase the intention of participants to use vasectomy. Similar studies conducted in Indonesia stated that as the number of alive children increases the intention to use vasectomy was also increased [20, 32]. Likewise, research conducted in India also concluded that the number of living children was a significant predictor for intention to use vasectomy [33]. The finding of this result was also supported by studies performed in Kenya and Rwanda [18, 19]. The possible reason may be men who had a

smaller number of children might have a high future fertility desire and their intention to use the contraceptive methods might be low. In other words, since vasectomy is permanent and irreversible, it is a choice of contraceptive method for those men who had more children and for those who want to limit their family.

The result of this study also showed that there was a positive relationship between knowledge of vasectomy and its acceptance by married men; it revealed that participants who had good knowledge about vasectomy had a higher intention to use it. The finding was also in line with a study conducted in Nigeria and the Kingdom of Eswatini [34] and [11] respectively. This might imply that knowledge plays an important role in the intention of vasectomy by increasing the awareness of individuals, having good knowledge about vasectomy used to helps men to know the importance of it from other methods avoids different misconceptions, changes the behavior and positively affects the attitude of men towards vasectomy.

Men who had a positive attitude towards vasectomy had increased intention to use it from that of men who had a negative attitude. The finding of this study was in agreement with the study finding from Nepal and Indonesia [21] and [20].

This might be attitude is a key factor that influence the intention, men with positive attitude towards vasectomy are better able to use it and share responsibilities in FP practice with their partner. The other reason may be individual who had positive attitude, they can break myths and misconception that was negatively affects intention to use vasectomy.

Generally, even though the health policy of Ethiopia puts or states different strategies to decrease maternal mortality like optimum utilization of family planning, but no practical change in attitudinal, behavioral, cultural and reliogional change is showed in men's family planning utilization like vasectomy; therefore, the countries policy should work for practical change rather theoretical report. In addition, for researcher community based, qualitative studies should run, that can find out more explorative factors.

## Conclusion

In conclusion, the prevalence of intention of married men to use vasectomy for future life was inlined with a study done in four regions of Ethiopia (Amhara, Oromia, SNNP and Tigray). Age (30–39) years, having more than three living children, having good knowledge, and a positive attitude towards vasectomy were significantly associated with the intention to use vasectomy for future life as a contraceptive method. As per finding, improving level of knowledge and attitude towards vasectomy is an essential strategy to scale up the intention of men to use vasectomy. We also recommend further researchers to come up with additional and detailed findings especially on qualitative aspect.

## Supporting information

**S1 File. English version questionnaires.**
(DOCX)

## Acknowledgments

The authors would like to acknowledge College of Medicine and Health Sciences, Bahir Dar University, for sponsoring this research project. We would also like to extend our heart full gratitude to Debre Tabor town Administration for permitting to conduct the study and providing the necessary preliminary information. The last but not the list we would like to extend our appreciation to the study participants, supervisor and data collectors.

## Author Contributions

**Conceptualization:** Alemu Degu Ayele, Fentahun Yenealem Beyene, Bekalu Getnet Kassa.

**Data curation:** Alemu Degu Ayele, Fentahun Yenealem Beyene, Kihinetu Gelaye Wudineh.

**Formal analysis:** Alemu Degu Ayele, Fentahun Yenealem Beyene, Kihinetu Gelaye Wudineh, Bekalu Getnet Kassa, Yitayal Ayalew Goshu, Gedefaye Nibret Mihretie.

**Funding acquisition:** Alemu Degu Ayele, Kihinetu Gelaye Wudineh, Gedefaye Nibret Mihretie.

**Investigation:** Alemu Degu Ayele, Fentahun Yenealem Beyene, Bekalu Getnet Kassa.

**Methodology:** Fentahun Yenealem Beyene, Kihinetu Gelaye Wudineh.

**Project administration:** Kihinetu Gelaye Wudineh, Yitayal Ayalew Goshu.

**Resources:** Kihinetu Gelaye Wudineh, Yitayal Ayalew Goshu.

**Software:** Fentahun Yenealem Beyene, Kihinetu Gelaye Wudineh.

**Supervision:** Kihinetu Gelaye Wudineh, Gedefaye Nibret Mihretie.

**Validation:** Fentahun Yenealem Beyene, Gedefaye Nibret Mihretie.

**Visualization:** Gedefaye Nibret Mihretie.

**Writing – original draft:** Fentahun Yenealem Beyene, Gedefaye Nibret Mihretie.

**Writing – review & editing:** Alemu Degu Ayele, Fentahun Yenealem Beyene, Bekalu Getnet Kassa.

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
