## [Decision Letter · Decision Letter 0]

4 Feb 2020

PONE-D-19-27356

Intention to use vasectomy and its associated factors among married men in Debretabor Town, North West Ethiopia, 2019.

PLOS ONE

Dear Mr. Ayele,

Thank you for submitting your manuscript to PLOS ONE. After careful consideration, we feel that it has merit but does not fully meet PLOS ONE’s publication criteria as it currently stands. Therefore, we invite you to submit a revised version of the manuscript that addresses the points raised during the review process.

We would appreciate receiving your revised manuscript by Mar 20 2020 11:59PM. To enhance the reproducibility of your results, we recommend that if applicable you deposit your laboratory protocols in protocols.io, where a protocol can be assigned its own identifier (DOI) such that it can be cited independently in the future. For instructions see: http://journals.plos.org/plosone/s/submission-guidelines#loc-laboratory-protocols

We look forward to receiving your revised manuscript.

Kind regards,

Sphiwe Madiba, DrPH

Academic Editor

PLOS ONE

2. Please include additional information regarding the survey or questionnaire used in the study and ensure that you have provided sufficient details that others could replicate the analyses. If you developed and/or translated a questionnaire as part of this study and it is not under a copyright more restrictive than CC-BY, please include a copy, in both the original language and English, as Supporting Information.

Additional Editor Comments

Please revise the introduction to provide a generalized background vasectomy and its utilization, benefits, family planning, involving men in reproductive health, factors associated with low uptake of vasectomy and opinions and attitudes of men about vasectomy. There is adequate and recent data on the topic in sub Saharan AfricaI am recommending the following articles;

Please read Shattuck et al, A review of 10 years of vasectomy programming and research in low-resource settings. Glob. Health: Sci. Pract. **2016**, 4, 647–660 for the global statistics on vasectomy,

Also read Shongwe et al, Assessing the Acceptability of Vasectomy as a Family

*Planning Option: A Qualitative Study with Men in the Kingdom of Eswatini, **nt. J. Environ. Res. Public Health****2019**, **16**, 5158. Please tell us about the factors associated with the low uptake of vasectomy *

Barriers to O_ering Vasectomy at Publicly Funded Family Planning Organizations in Texas. Am. J. Men Health **2017**,

Please read, Kısa et al. Opinions and attitudes about vasectomy of married couples living in Turkey. Am. J. Men Health **2017**, 11, 531–541

Mar, S.O.; Ali, O.; Sandheep, S.; Husayni, Z.; Zuhri, M. Attitudes towards vasectomy and its acceptance as a method of contraception among clinical-year medical students in a Malaysian private medical college

**Methods**

Please follow the journal’s standard subheadings for the methodology.Integrate the operational definitions with the measures and data analysis, write as a narrative, and remove the bolding from the words.

**Data collection**

Provide details on the tool and type of questions that were asked. Was the tool validated, how was intention measured-the type of question, what does future intention mean- is there a time factor on the future?

**Results**

Please follow the standardized categories for age and reanalyse the data accordingly, correct for all the tables and narratives in the document.

**Discussion **

The discussion need a revision in line with the introductionPlease tell us about those who intended to use vasectomy-what were their demographics before you do the comparisons with other studies, first in Ethiopia and other countriesYou spend too much effort of why your study findings are different from those of others and not on the implications of your findings.

**Conclusion**

Improving level of knowledge and attitude towards vasectomy is an essential strategy-details on how this will be doneThe document should be copy edited for language and grammar by a professional language editor before submitting your revised manuscript.  

Reviewers' comments:

Reviewer's Responses to Questions

**Comments to the Author**

1. Is the manuscript technically sound, and do the data support the conclusions?

Reviewer #1: Partly

Reviewer #2: No

2. Has the statistical analysis been performed appropriately and rigorously? 

Reviewer #1: I Don't Know

Reviewer #2: No

3. Have the authors made all data underlying the findings in their manuscript fully available?

Reviewer #1: No

Reviewer #2: Yes

4. Is the manuscript presented in an intelligible fashion and written in standard English?

Reviewer #1: No

Reviewer #2: No

5. Review Comments to the Author

Reviewer #1: This paper addresses an important topic – vasectomy and whether men would be willing to use the method and the characteristics of men expressing such willingness. The title of the manuscript is misleading = it says “intention” but the variable in the paper is “willingness” – those are not the same thing. The authors should be clear what the men were asked out (intention or willingness – and say in the paper the question/questions asked about willingness or intention to use.

Knowledge of vasectomy: The authors say that knowledge of vasectomy was measured through 9 questions – but do not say what those questions are. The questions should be included in the paper.

Attitudes about vasectomy: It is not clear how knowledge scores can translate into positive or negative attitudes towards vasectomy. One could have good knowledge and still have a negative attitude about the method.

What does “emotional support” from their partners to use vasectomy mean? How was this question asked? Were the men asked about current contraceptive use by themselves or their wives/partners?

Willingness (or intention?) to use: The authors should explain whey they did not ask the men if they knew where they could get a vasectomy and what would prompt them to actually have a vasectomy in the future.

Furthermore, the findings of willingness or intention need to be put in the context of the availability of vasectomy in the country. Without available services, men will not be able to act on their willingness or intentions.

The current comparison of the findings with findings from studies in other countries is interesting, but ultimately without expanded services, knowledge and willingness (or intention) is not sufficient. The authors could use the findings that are fairly consistent across studies to say that their seems to be latent interest in vasectomy that is not being satisfied with programming.

The authors should note that the sample in this study are highly educated (63% have a college education or above, and half the wives also have a college education or above) – what is the percent of the population in Ethiopia with a college education? The authors should explain why the town of Debre Tabor was selected for the study.

With that said, the men in this study could be vanguard users of vasectomy, so the findings are useful for advocating for expansion of vasectomy services in Ethiopia. It would be useful for the authors to refer to Ethiopia’s FP Costed Implementation Plan 2016-2020 to see if vasectomy is covered in it (http://www.healthpolicyplus.com/ns/pubs/2021-2030_EthiopiaCIPNov.pdf). FHI360 has prepared a brief on promoting vasectomy in Ethiopia that also might be helpful (https://www.fhi360.org/sites/default/files/media/documents/resource-vasectomy-evidence-ethiopia-final.pdf).

References should be listed by Last Name, First name (currently first names and last initial are included in the references).

This paper would benefit from a more thorough literature review on vasectomy and programming for men. Some other references to consider:

Vasectomy: A Long, Slow Haul to Successful Takeoff, James D Shelton and Roy Jacobstein, Global Health: Science and Practice December 2016, 4(4):514-517; https://doi.org/10.9745/GHSP-D-16-00355 Glob Health Sci Pract. 2016 Dec 23; 4(4): 647–660.

A Review of 10 Years of Vasectomy Programming and Research in Low-Resource Settings, Dominick Shattuck, Brian Perry, Catherine Packer, and Dawn Chin Quee , Global Health: Science and Practice, Published online 2016 Dec 23. doi: 10.9745/GHSP-D-16-00235

Hardee, K, M Croce-Galis, and J Gay. 2017. “Are Men Well Served by Family Planning Programs?” Reproductive Health. 14(14). DOI: 10.1186/s12978-017-0278-5.

Ross, J, and K Hardee. 2016. “Use of Male Methods of Contraception Worldwide.” 2016. Journal of Biosocial Science. Published online.

Reviewer #2: Even though the topic is of public health importance as we strive to promote the use of male methods in the family planning method mix, the manuscript as it is now is very poorly written. It is extremely difficult to understand the content because of the poor quality of English. Additionally, the sampling procedure is inadequately explained - for example how was the first household selected in each kebele and how was the sample size for each kebele arrived at? It is also unclear how 'the next eligible household located in the clockwise direction was visited..' Another issue is that the authors did not provide an operational for 'intention to use vasectomy' and the Table titles are inappropriate.

The manuscript as it stands now will require major revisions, and should be seen by an English proof-reader before re-submission

6. PLOS authors have the option to publish the peer review history of their article (what does this mean?). If published, this will include your full peer review and any attached files.

Reviewer #1: Yes: Karen Hardee

Reviewer #2: Yes: Easmon Otupiri

---

## [Author Response · Author response to Decision Letter 0]

11 Mar 2020

To Plose One Academic Editor (Sphiwe Madiba, DrPH). 

PONE-D-19-27356

From Alemu Degu

First of all I would like to say thank you very much for your timely response and for your interesting comments. And my heart full gratitude to all reviewers those giving essential comments and questions by devoting their golden time.

Point by point response to editors 

I. General comments 

1. Please ensure that your manuscript meets PLOS ONE's style requirements, including those for file naming. The PLOS ONE style templates can be foundathttp://www.journals.plos.org/plosone/s/file?id=wjVg/PLOSOne_formatting_sample_main_body.pdfandhttp://www.journals.plos.org/plosone/s/file?id=ba62/PLOSOne_formatting_sample_title_authors_affiliations.pdf

Response 1: I accept the comment and I have fully follows the journals quid line (style requirements).

2. Please include additional information regarding the survey or questionnaire used in the study and ensure that you have provided sufficient details that others could replicate the analyses. If you developed and/or translated a questionnaire as part of this study and it is not under a copyright more restrictive than CC-BY, please include a copy, in both the original language and English, as Supporting Information.

Response 2: I accept the comment and I have incorporated the questionnaire as additional information.

II specific comments 

A. On introduction

1. Please revise the introduction to provide a generalized background vasectomy and its utilization, benefits, family planning, involving men in reproductive health, factors associated with low uptake of vasectomy and opinions and attitudes of men about vasectomy. There is adequate and recent data on the topic in sub Saharan Africa

Response: I accept the comment and I have incorporated and rewrite the introduction and discussion part of the main document. 

B. Methods

1. Please follow the journal’s standard subheadings for the methodology.

2. Integrate the operational definitions with the measures and data analysis, write as a narrative, and remove the bolding from the words.

Response: I accept the comment and I have followed the journals guideline in all aspects. 

C. Data collection

1. Provide details on the tool and type of questions that were asked. Was the tool validated, how was intention measured-the type of question, what does future intention mean- is there a time factor on the future?

Response: I accept the comment and I have included in detail of the tool and type of questions. The tool was validated by those individuals who have experienced in related subject matter and future intention means view, thinking or expectation to use /yes there is time factor. See on the manuscript result part. 

D. Results

1. Please follow the standardized categories for age and reanalyse the data accordingly, correct for all the tables and narratives in the document.

Response: I accept the comment, but I have consider the previous related research age categories for comparison and discussion purpose and no participant age <20 years at the time of data collection, since the number of respondents are small those are aged in between 50 -59, 60-69, 70-79...merged as age >=50 years. 

D. Discussion 

1. The discussion need a revision in line with the introduction

2. Please tell us about those who intended to use vasectomy-what were their demographics before you do the comparisons with other studies, first in Ethiopia and other countries

3. You spend too much effort of why your study findings are different from those of others and not on the implications of your findings.

Response: I accept the comment and incorporated to the main document (manuscript) based on your comments. 

E. Conclusion

1. Improving level of knowledge and attitude towards vasectomy is an essential strategy-detail on how this will be done

2. The document should be copy edited for language and grammar by a professional language editor before submitting your revised manuscript. 

Response: I accept the comment and the question; we can improve the level of knowledge and attitude by giving detail awareness on the methods and benefit of it and by avoiding false remorse that talking about vasectomy. We have seen the language we ourselves the author +online grammar checker +by professional language editor.

II. Point by point response to Reviewers

For Reviewer#1

 #1: This paper addresses an important topic – vasectomy and whether men would be willing to use the method and the characteristics of men expressing such willingness. The title of the manuscript is misleading = it says “intention” but the variable in the paper is “willingness” – those are not the same thing. The authors should be clear what the men were asked out (intention or willingness – and say in the paper the question/questions asked about willingness or intention to use.

Response#1: I accept the comment and the question widely. Yes all write, Intention and willingness is not the same thing. Where taking to Intention it is mostly explains futures expectation, thinking but the possibility may or may not. Whereas willingness is being willing to do that thing on the time or future. Generally Intention is some body’s view, thinking, expectation to the future and it is somewhat light /superficial whereas willingness is some bodies willing to act/ do or to perform and it is somewhat hard. 

#2: Knowledge of vasectomy: The authors say that knowledge of vasectomy was measured through 9 questions – but do not say what those questions are. The questions should be included in the paper.

Attitudes about vasectomy: It is not clear how knowledge scores can translate into positive or negative attitudes towards vasectomy. One could have good knowledge and still have a negative attitude about the method. What does “emotional support” from their partners to use vasectomy mean? How was this question asked? Were the men asked about current contraceptive use by themselves or their wives/partners?

Response#2: I accept the comment and the question and we have included questions that we used to measure Knowledge of vasectomy in the edited manuscript. Yes write that those having good Knowledge of vasectomy may have a negative attitude about the method. And there are attitude measurement questions and knowledge measurement questions separately. ‘Emotional support’ to mean that; have you got a support or volunteer from wives, I think we use emotional inappropriate word selection and make the confusion. And we are going to asking like this “Do you get support from your partner to use FP?” Currently we accept the comment and corrected like this. Do you get support from your partner to use FP?”

#3. Willingness (or intention?) to use: The authors should explain whey they did not ask the men if they knew where they could get a vasectomy and what would prompt them to actually have a vasectomy in the future.

 Response #3: I accept the comment and the question, but the question is not clear ; in my understanding to mean that place of vasectomy ; And we are asking the availability of the service like this “Do you know where vasectomy/male sterilization service is available? So place of vasectomy is answered by this question.

 #4: Furthermore, the findings of willingness or intention need to be put in the context of the availability of vasectomy in the country. Without available services, men will not be able to act on their willingness or intentions.

Response #4: I accept the comment and the availability of the service is largely there. In any district hospital, general hospital can be performed. And the service also given in the study area. 

#5; The current comparison of the findings with findings from studies in other countries is interesting, but ultimately without expanded services, knowledge and willingness (or intention) is not sufficient. The authors could use the findings that are fairly consistent across studies to say that their seems to be latent interest in vasectomy that is not being satisfied with programming.

Response #4: I accept the comment and the service is at large present in our country. But insufficient number of users is there?

#5. The authors should note that the sample in this study are highly educated (63% have a college education or above, and half the wives also have a college education or above) – what is the percent of the population in Ethiopia with a college education? The authors should explain why the town of Debre Tabor was selected for the study.

Response #5: I accept the comment and questions ; the sample in our study is highly educated which is 63% ; the reason is that the study conducted is in the town and small place and may easily accessible to education whereas the general population educational level in Ethiopia is less than this is due to the above explanation . The reason I select the study area is ; since am working there , I am not still see the procedure of vasectomy even the service is available . Due to this fact I intended to assess the intention and its factors in this area. 

#6; With that said, the men in this study could be vanguard users of vasectomy, so the findings are useful for advocating for expansion of vasectomy services in Ethiopia. It would be useful for the authors to refer to Ethiopia’s FP Costed Implementation Plan 2016-2020 to see if vasectomy is covered in it (http://www.healthpolicyplus.com/ns/pubs/2021-2030_EthiopiaCIPNov.pdf). FHI360 has prepared a brief on promoting vasectomy in Ethiopia that also might be helpful (https://www.fhi360.org/sites/default/files/media/documents/resource-vasectomy-evidence-ethiopia-final.pdf).

References should be listed by Last Name, First name (currently first names and last initial are included in the references).

This paper would benefit from a more thorough literature review on vasectomy and programming for men. Some other references to consider:

Vasectomy: A Long, Slow Haul to Successful Takeoff, James D Shelton and Roy Jacobstein, Global Health: Science and Practice December 2016, 4(4):514-517; https://doi.org/10.9745/GHSP-D-16-00355 Glob Health Sci Pract. 2016 Dec 23; 4(4): 647–660.

A Review of 10 Years of Vasectomy Programming and Research in Low-Resource Settings, Dominick Shattuck, Brian Perry, Catherine Packer, and Dawn Chin Quee , Global Health: Science and Practice, Published online 2016 Dec 23. doi: 10.9745/GHSP-D-16-00235

Hardee, K, M Croce-Galis, and J Gay. 2017. “Are Men Well Served by Family Planning Programs?” Reproductive Health. 14(14). DOI: 10.1186/s12978-017-0278-5.

Ross, J, and K Hardee. 2016. “Use of Male Methods of Contraception Worldwide.” 2016. Journal of Biosocial Science. Published online.

Response #5: I accept the comment and I have revised the literature based on your comment and suggestions.

Thank you !!!.

Alemu Degu

For Reviewer#2

Even though the topic is of public health importance as we strive to promote the use of male methods in the family planning method mix, the manuscript as it is now is very poorly written. It is extremely difficult to understand the content because of the poor quality of English. Additionally, the sampling procedure is inadequately explained - for example how the first household was selected in each kebele and how was the sample size for each kebele arrived at? It is also unclear how 'the next eligible household located in the clockwise direction was visited..' Another issue is that the authors did not provide an operational for 'intention to use vasectomy' and the Table titles are inappropriate.

The manuscript as it stands now will require major revisions, and should be seen by an English proof-reader before re-submission

Response: I accept the comment and questions; I have taken the comment as a comment and corrected the manuscript based on your comment. Largely the issue is on grammar and language; I accept it edited at much as possible with online grammar and language checkers and with my friend fluent English speaker in my university. The first house hold is selected rondomely by lottery method for each kebele, after the number of households with married men in each kebele were found from the kebele registration book. Then the study households were selected from each kebele through simple random sampling technique by using computer generating table starting from kebele one from a random start point. One married man per household was interviewed. See it again I have operationalized it . The table title “ I have corrected based on the journal guideline. Thank you!!!.

Alemu Degu

---

## [Decision Letter · Decision Letter 1]

27 Apr 2020

PONE-D-19-27356R1

Intention to use vasectomy and its associated factors among married men in Debretabor Town, North West Ethiopia, 2019.

PLOS ONE

Dear Mr. Ayele,

Thank you for submitting your manuscript to PLOS ONE. After careful consideration, we feel that it has merit but does not fully meet PLOS ONE’s publication criteria as it currently stands. Therefore, we invite you to submit a revised version of the manuscript that addresses the points raised during the review process.

We would appreciate receiving your revised manuscript by Jun 11 2020 11:59PM. To enhance the reproducibility of your results, we recommend that if applicable you deposit your laboratory protocols in protocols.io, where a protocol can be assigned its own identifier (DOI) such that it can be cited independently in the future. For instructions see: http://journals.plos.org/plosone/s/submission-guidelines#loc-laboratory-protocols

We look forward to receiving your revised manuscript.

Kind regards,

Wen-Jun Tu

Academic Editor

PLOS ONE

Reviewers' comments:

Reviewer's Responses to Questions

**Comments to the Author**

1. If the authors have adequately addressed your comments raised in a previous round of review and you feel that this manuscript is now acceptable for publication, you may indicate that here to bypass the “Comments to the Author” section, enter your conflict of interest statement in the “Confidential to Editor” section, and submit your "Accept" recommendation.

Reviewer #1: (No Response)

2. Is the manuscript technically sound, and do the data support the conclusions?

Reviewer #1: Partly

3. Has the statistical analysis been performed appropriately and rigorously? 

Reviewer #1: Yes

4. Have the authors made all data underlying the findings in their manuscript fully available?

Reviewer #1: Yes

5. Is the manuscript presented in an intelligible fashion and written in standard English?

Reviewer #1: No

6. Review Comments to the Author

Reviewer #1: The authors have addressed the reviewers’ comments on the original manuscript for the most part. A few issues are still outstanding. The authors describe intention to use vasectomy as “willingness of respondent to use vasectomy as a contraceptive method for future time.” This mixed intention and willingness, which are not the same things. Intention is stronger than willingness and implies that the man is planning to use vasectomy. Willingness implies that he might be interested in using it at some time in the future. The authors need to be clear what was asked – otherwise this seems like a vague outcome that might be more due to respondent courtesy bias than actual interest in using the method.

Is there a reason the men were not asked if they (their wives) were currently using a contraceptive method? Most of the men said they had completed their family size, so it would be important to know if they / their wives were currently doing something to avoid pregnancy.

Figure 2 is labeled “reason not to use vasectomy for future life” – is that how the question on intention to use was asked? This is likely a translation issue – what does it mean to use a method for future life?

Figure 2 shows that 74% said the reason not to use vasectomy was lack of awareness – yet 46.6% said they had heard of vasectomy according to Table 3. If nearly half of the men said they had heard of vasectomy, how could three-quarters say unfamiliarity with the method was a reason not to use it?

The discussion would be strengthened if it focused less on how the findings of the study match other studies and more on the programmatic implications for Ethiopia – how can the findings from this study improve access to and use of vasectomy in Ethiopia.

The manuscript still needs English language editing.

7. PLOS authors have the option to publish the peer review history of their article (what does this mean?). If published, this will include your full peer review and any attached files.

Reviewer #1: No

---

## [Author Response · Author response to Decision Letter 1]

21 May 2020

To Plose One Academic Editor 

PONE-D-19-27356R1

From Alemu Degu

First of all I would like to say thank you very much for your timely response and your actively involvement on the comments and coordination without hesitating by the current issue (COVID 19). And my heart full gratitude again reaches to all reviewers those giving a measurable, fruitful and observable comments and questions by devoting their golden time and with the challenge of COVID19. I am happy to see you the next time and year. God bless to you and us!!! 

I. Point by point response to Reviewers

For Reviewer#1

1. The authors describe intention to use vasectomy as “willingness of respondent to use vasectomy as a contraceptive method for future time.” This mixed intention and willingness, which are not the same things. Intention is stronger than willingness and implies that the man is planning to use vasectomy. Willingness implies that he might be interested in using it at some time in the future. The authors need to be clear what was asked – otherwise this seems like a vague outcome that might be more due to respondent courtesy bias than actual interest in using the method

Response#1: We accept the comment and the question. Yes all write, Intention and willingness is not the same thing, even though some scholars give they are synonym at some point of time. As we have searched and reading different articles and different scholars point of view “Intension to use something” is the view of futurity, planning to use in the future, their future motivation to use and their effort to use in the future whereas “willingness to use something” is being willing to do that thing on the time, readiness to use and openness to perform or act. Therefore based on these and our consensus we have tried to assess their intension to use vasectomy as a contraceptive method for future time rather their willingness. We have tried to ask the respondents on part VI question number 501and 502”Do you have intention to use vasectomy for future? If yes when? “; on these we want to assess their future planning or view rather willingness. We can directly interpret the above question like this one “Do you have plan to use vasectomy for future? If yes when? “

And also we have assessed published articles related to this topic and the variables were adapted from previously published articles.

2. Is there a reason the men were not asked if they (their wives) were currently using a contraceptive method? Most of the men said they had completed their family size, so it would be important to know if they / their wives were currently doing something to avoid pregnancy.

Response#2: We accept the comment and the question. This is a very easy but serious issue. We are making some systematic error that makes the authors confused. Around 281 respondents had completed their family size and 116 respondents have not completed. We have tried to ask 116 respondents on future desire no of children but wrongly we report 281 respondents, we make a mistake on copy paste to the table. Those 281 respondents no need of asking about their future desire no of children, since they side already completed family size. Generally those men who have (their wives) were currently using a contraceptive method were asked about future intention to use vasectomy as contraceptive method unless their wives used permanent methods like bilateral tubal ligation.. or both those complete their family size and not complete were asked about future intention to use vasectomy as contraceptive method. We have corrected on the manuscript. 

3. Figure 2 is labeled “reason not to use vasectomy for future life” – is that how the question on intention to use was asked? This is likely a translation issue – what does it mean to use a method for future life?

Response#3: We accept the comment and the question. Yes it makes some confusion on word utilization. And we have corrected as “Percentage distribution of study participants by reason not have intention to use vasectomy for future life ; Debre Tabor Town, North West Ethiopia, March 05-April 15 2019”.

4. Figure 2 shows that 74% said the reason not to use vasectomy was lack of awareness – yet 46.6% said they had heard of vasectomy according to Table 3. If nearly half of the men said they had heard of vasectomy, how could three-quarters say unfamiliarity with the method was a reason not to use it?

Response#4: We accept the comment and the question. To avoid these confusion better to describe difference between awareness and simply heard about vasectomy. Having awareness about vasectomy is that having detail knowledge of vasectomy about its purpose, procedure, side effect and advantage whereas heard about vasectomy is simply hear the presence of vasectomy not have detail knowledge. Therefore those having heard about vasectomy might or might not have awareness but those have awareness were heard about vasectomy. Therefore even though 46.6 % of respondents heard about vasectomy, they may or may not have awareness. Mostly hearing information is easy but when we asking how, for home, when…Is not answered by the respondents.

5. The discussion would be strengthened if it focused less on how the findings of the study match other studies and more on the programmatic implications for Ethiopia – how the findings from this study can improve access to and use of vasectomy in Ethiopia.

Response#4: We accept the comment and the question. And we have tried to edit based on the comment.

6. The manuscript still needs English language editing.

Response#4: We accept the comment and the question. And we have tried to edit as much as possible with online grammar and language checkers and with my friend fluent English speaker in my university.

 Thank you!!!

Alemu Degu

---

## [Decision Letter · Decision Letter 2]

27 May 2020

PONE-D-19-27356R2

Intention to use vasectomy and its associated factors among married men in Debretabor Town, North West Ethiopia, 2019.

PLOS ONE

Dear Dr. Ayele,

Thank you for submitting your manuscript to PLOS ONE. After careful consideration, we feel that it has merit but does not fully meet PLOS ONE’s publication criteria as it currently stands. Therefore, we invite you to submit a revised version of the manuscript that addresses the points raised during the review process.

We look forward to receiving your revised manuscript.

Kind regards,

Wen-Jun Tu

Academic Editor

PLOS ONE

Reviewers' comments:

Reviewer's Responses to Questions

**Comments to the Author**

1. If the authors have adequately addressed your comments raised in a previous round of review and you feel that this manuscript is now acceptable for publication, you may indicate that here to bypass the “Comments to the Author” section, enter your conflict of interest statement in the “Confidential to Editor” section, and submit your "Accept" recommendation.

Reviewer #1: (No Response)

2. Is the manuscript technically sound, and do the data support the conclusions?

Reviewer #1: Partly

3. Has the statistical analysis been performed appropriately and rigorously? 

Reviewer #1: No

4. Have the authors made all data underlying the findings in their manuscript fully available?

Reviewer #1: Yes

5. Is the manuscript presented in an intelligible fashion and written in standard English?

Reviewer #1: No

6. Review Comments to the Author

Reviewer #1: This version of the manuscript has not addressed completely the comments I made on the second revision and I’ve noticed some additional issues.

Table 3 shows that 53.4% of the men had NOT heard about vasectomy. I should have caught this the first time I reviewed the paper. These men should have been skipped out of responding to other questions about vasectomy – if they haven’t heard of the method, how can they respond to specific questions about the method? The rest of the analysis should only be done on the 46.6% of the men who had heard of vasectomy. The conclusion that lack of knowledge about the method is an impediment to expanding use of vasectomy is still important. And then knowing what men with knowledge of the method think about it and their intention of using it will be programmatically useful.

Also, something I noticed rereading the paper – what does it mean in Table 2 that 74% said they got emotional support from their partners to use vasectomy? Again, how could they get support from their partners that they haven’t heard of?

The discussion section is still weak – it still focuses too much on comparison of findings from Ethiopia and other countries and not enough on how the findings are useful for Ethiopia’s program. I am recopying my previous comments:

I had previously suggested that the authors put the findings in the context that the sample is highly educated - 63% have a college education or above, and half the wives also have a college education or above – which is higher than Ethiopia’s average educational attainment. The men in this study could be vanguard users of vasectomy, so the findings are useful for advocating for expansion of vasectomy services in Ethiopia. It would be useful for the authors to refer to Ethiopia’s FP Costed Implementation Plan 2016-2020 to see if vasectomy is covered in it (http://www.healthpolicyplus.com/ns/pubs/2021-2030_EthiopiaCIPNov.pdf). FHI360 has prepared a brief on promoting vasectomy in Ethiopia that also might be helpful (https://www.fhi360.org/sites/default/files/media/documents/resource-vasectomy-evidence-ethiopia-final.pdf).

In my first review, I had suggested looking at additional references on vasectomy, which I note the authors have not included in the references, so it appears they did not review them. My comment still stands that this paper would benefit from a more thorough literature review on vasectomy and programming for men. For example,

Vasectomy: A Long, Slow Haul to Successful Takeoff, James D Shelton and Roy Jacobstein, Global Health: Science and Practice December 2016, 4(4):514-517; https://doi.org/10.9745/GHSP-D-16-00355 Glob Health Sci Pract. 2016 Dec 23; 4(4): 647–660.

A Review of 10 Years of Vasectomy Programming and Research in Low-Resource Settings, Dominick Shattuck, Brian Perry, Catherine Packer, and Dawn Chin Quee , Global Health: Science and Practice, Published online 2016 Dec 23. doi: 10.9745/GHSP-D-16-00235

Hardee, K, M Croce-Galis, and J Gay. 2017. “Are Men Well Served by Family Planning Programs?” Reproductive Health. 14(14). DOI: 10.1186/s12978-017-0278-5.

Ross, J, and K Hardee. 2016. “Use of Male Methods of Contraception Worldwide.” 2016. Journal of Biosocial Science. Published online.

Also, while the English is better, the manuscript still needs editing for English language usage.

7. PLOS authors have the option to publish the peer review history of their article (what does this mean?). If published, this will include your full peer review and any attached files.

Reviewer #1: No

---

## [Author Response · Author response to Decision Letter 2]

15 Jun 2020

To Plose One Academic Editor 

PONE-D-19-27356R2

From Fentahun Yenealem

First of all I would like to say thank you very much for your timely response and your actively involvement on the comments and coordination without hesitating by the current issue (COVID 19). And my heart full gratitude again reaches to all reviewers those giving a measurable, fruitful and observable comments and questions by devoting their golden time and with the challenge of COVID19. 

I. Point by point response to Reviewers

For Reviewer#1

Q1. Table 3 shows that 53.4% of the men had NOT heard about vasectomy. I should have caught this the first time I reviewed the paper. These men should have been skipped out of responding to other questions about vasectomy – if they haven’t heard of the method, how can they respond to specific questions about the method? The rest of the analysis should only be done on the 46.6% of the men who had heard of vasectomy. The conclusion that lack of knowledge about the method is an impediment to expanding use of vasectomy is still important. And then knowing what men with knowledge of the method think about it and their intention of using it will be programmatically useful.

Response#1: We accept the comment and the question. Yes alright, we have made a great mistake on using language /taking or writing the manuscript rather data collection; we using Heard about vasectomy to mean or replaces that Do you know that vasectomy is a contraceptive method by cutting and ligating Vass deference? So this is assessing the knowledge of men’s about its procedure. Yes you are right in previous situation completely it is skip question; but wrongly usage of language. If you have time it is possible to see and check its consistency in additional file 1. We have made correction based on the comment both track change and clear manuscript. 

Q2. Also, something I noticed rereading the paper – what does it mean in Table 2 that 74% said they got emotional support from their partners to use vasectomy? Again, how could they get support from their partners that they haven’t heard of?

Response#2: We accept the comment and the question. Yes alright, the question might be answered in the above response since heard of vasectomy is corrected accordingly.

Q#3: The discussion would be strengthened if it focused less on how the findings of the study match other studies and more on the programmatic implications for Ethiopia – how the findings from this study can improve access to and use of vasectomy in Ethiopia.

Response#4: We accept the comment and the question. And we have tried to edit based on the comment and based on your recommendation.

Q#4.The manuscript still needs English language editing.

Response#4: We accept the comment and the question. And we have tried to edit as much as possible with online grammar and language checkers and with my friend fluent English speaker in my university.

 Thank you!!!

Fentahun Yenealem

---

## [Decision Letter · Decision Letter 3]

19 Jun 2020

PONE-D-19-27356R3

Intention to use vasectomy and its associated factors among married men in Debretabor Town, North West Ethiopia, 2019.

PLOS ONE

Dear Dr. Beyene,

Thank you for submitting your manuscript to PLOS ONE. After careful consideration, we feel that it has merit but does not fully meet PLOS ONE’s publication criteria as it currently stands. Therefore, we invite you to submit a revised version of the manuscript that addresses the points raised during the review process.

We look forward to receiving your revised manuscript.

Kind regards,

Wen-Jun Tu

Academic Editor

PLOS ONE

Reviewers' comments:

Reviewer's Responses to Questions

**Comments to the Author**

1. If the authors have adequately addressed your comments raised in a previous round of review and you feel that this manuscript is now acceptable for publication, you may indicate that here to bypass the “Comments to the Author” section, enter your conflict of interest statement in the “Confidential to Editor” section, and submit your "Accept" recommendation.

Reviewer #3: (No Response)

2. Is the manuscript technically sound, and do the data support the conclusions?

Reviewer #3: Yes

3. Has the statistical analysis been performed appropriately and rigorously? 

Reviewer #3: Yes

4. Have the authors made all data underlying the findings in their manuscript fully available?

Reviewer #3: Yes

5. Is the manuscript presented in an intelligible fashion and written in standard English?

Reviewer #3: No

6. Review Comments to the Author

Reviewer #3: This is a valuable empirical contribution to an understanding of men's potential use of vasectomy, in Ethiopia but in international scope too. I did not review earlier version(s) of the manuscript. My overall sense is that the research design is satisfactory, that the findings follow from the design, and that the key findings stand out clearly.

My major concern is that the writing quality is inadequate for the journal, and that a more professional editorial process is necessary before a final version can be accepted. To illustrate some examples of spelling or grammatical concerns that are of scientific merit in how the findings are reported, Vass difference should be vas deferens; alive children would be better phrased living children; and I atttempt my own version of the Abstract and Key Words (cut and pasted below) that I think illustrates what would be a more polished version than the current one. In a related vein, the formatting of the Tables and Figure needs to be improved (e.g., columns don't align), and any acronymy (e.g., EDHS) should be written out the first time introduced.

I also suggest several additions to the Introduction and Discussion as follows. a) Please discuss in the Discussion the high educational attainment and prevalent civil servant occupations of this sample: how do these compare with national Ethiopian data (perhaps compare with EDHS data, if available), and how generalizable or not are the current data? b) Perhaps also note what percent of men have 0, 1, 2 and 3 children just to share empirical data within the category of men having 3 or fewer children. c) In the Introduction when presenting data on Ethiopian family planning patterns, please also share some information about the prevalence of specific types of family planning such as condoms, tubal ligation, etc. d) In a related vein, in the Discussion, might another factor relevant to why men 40+ have lower vasectomy intent than men 30-39 be age- and fertility-related patterning of their wives' family planning (e.g., do older men's wives more often use tubal ligation, and does that mean that men's vasectomy decisions are somewhat contingent upon those of their wives). e) Please state the age range of male participants, in the Abstract and Results. I am guessing that these are men aged 20 years and older, but whether men in their teens were excluded was not noted in the Methods, nor is the upper limit of male ages specified.; f) in the Methods section, please craft a clearer sentence that specifies what the cutoffs are for differentiating good and poor knowledge. g) Does the item "Get emotional support from partner" refer to emotional support generally, or specifically to family planning support?

Background: Vasectomy is one of the most effective and permanent male contraceptive methods, and involves cutting and ligating the vas deferens to make the semen free of sperm during ejaculation. Although it is effective, simple and safe, it is not well known and practiced in the majority of our community. This study assessed intention to use vasectomy and its associated factors among married men in Debre Tabor Town, North West Ethiopia, 2019.

Methods: A community based cross-sectional study was conducted among 402 married men from March 05 to April 15, 2019. Simple random sampling technique was employed to select the study participants. Data were collected by face-to-face interview using a structured and pre-tested questionnaire. Questions concerned sociodemographic and reproductive variables and views on vasectomy. The association between variables was analyzed using bivariable and multivariable logistic regression model.

Results: A total of 402 participants were included with response rate of 98.75%. The mean participant age was 37.12 (SD ± 6.55) years. The prevalence of intention to use vasectomy was 19.6% with 95%CI (15.6%-23.4%). Multivariable logistic regression showed that age from 30-39 years (AOR=3.2 (95% CI: 1.19-8.86)), having more than three living children (AOR=2.5 (95% CI: 1.41-4.68)), good knowledge of vasectomy (AOR=3.4 (95%CI: 1.88-6.40)) and positive attitude toward vasectomy (AOR=4.8 (95% CI: 2.61-8.80)) were significantly associated with intention to use vasectomy.

Conclusion and recommendation: Intention to use vasectomy was comparable with findings in four regions of Ethiopia (Amhara, Oromia, SNNP and Tigray). Age, number of living children, knowledge and attitude were significantly associated with the intention to use vasectomy. Improving level of knowledge and attitude towards vasectomy is an essential strategy to scale up intention of men to use vasectomy.

Key words : Vasectomy, male contraception, family planning, fertility, Ethiopia

7. PLOS authors have the option to publish the peer review history of their article (what does this mean?). If published, this will include your full peer review and any attached files.

Reviewer #3: No

---

## [Author Response · Author response to Decision Letter 3]

22 Jul 2020

To Plose One Academic Editor 

PONE-D-19-27356R3

From Fentahun Yenealem

First of all I would like to say thank you very much for your timely response and your actively involvement on the comments and coordination without hesitating by the current issue (COVID 19). And my heart full gratitude again reaches to reviewer#3 that was giving a measurable, fruitful and observable comments and questions by devoting their golden time and with the challenge of COVID19. 

I. Point by point response to Reviewer

For Reviewer#3

General comments and recommendation: We have completely accepted the general comments and recommendations especially on the abstract section. Thank you very much for your recommendation!. Following this; We have tried to give response for each comments and questions raised by the reviewer.

Q #1 Please discusses in the Discussion the high educational attainment and prevalent civil servant occupations of this sample: how do these compare with national Ethiopian data (perhaps compare with EDHS data, if available), and how generalizable or not are the current data?

Response#1: We accept the comment and the question. We have made correction based on the comment both in track change and clear manuscript. We have tried to discussing with EDHS 2016 finding the educational attainment and civil servant occupations.

Q #2. Perhaps also note what percent of men have 0, 1, 2 and 3 children just to share empirical data within the category of men having 3 or fewer children.

Response#2: We accept the comment and the question. Yes alright, the question might be answered in the above response since heard of vasectomy is corrected accordingly. The total numbers of participants those having <= 3 children is 243. The empirical data without categorized to <=3 child. 1. Those have no child 25, having one child 67, having two children 113 and having three children 38. But on the manuscript we have write by category<=3 and> 3 child.

Q #3. In the Introduction when presenting data on Ethiopian family planning patterns, please also share some information about the prevalence of specific types of family planning such as condoms, , etc. 

Response#3: We accept the comment and the question. And we have tried to edit based on the comment and based on your recommendation. See in the clear manuscript and track changes.

Q#4. In a related vein, in the Discussion, might another factor relevant to why men 40+ have lower vasectomy intent than men 30-39 be age- and fertility-related patterning of their wives' family planning (e.g., do older men's wives more often use tubal ligation, and does that mean that men's vasectomy decisions are somewhat contingent upon those of their wives).

Response#4: We accept the comment and the question. And we have tried corporate based on your recommendation and question. 

Q#5. Please state the age range of male participants, in the Abstract and Results. I am guessing that these are men aged 20 years and older, but whether men in their teens were excluded was not noted in the Methods, nor is the upper limit of male ages specified.

Response#5: We accept the comment and the question. And we have tried to edit based on the comment and based on your recommendation. We are not excluded those teenager participants. But on interview no participants <20 and >56 years old. That is why we are not describing in the method section. And we have made correction based on your comment on the age range in between 20-56 years. See in the clear manuscript and track changes.

Q#6. in the Methods section, please craft a clearer sentence that specifies what the cutoffs are for differentiating good and poor knowledge.

Response#6: We accept the comment, recommendation and the question. But we have been describing it in the method section specifically in the sub heading of Measurement and data collection procedure (line 11-13).

Q#7: Does the item "Get emotional support from partner" refer to emotional support generally, or specifically to family planning support?

Response#7: We accept the comment, recommendation and the question. In this situation get emotional support from partner" refer to emotional support specifically to family planning support.

8# My major concern is that the writing quality is inadequate for the journal, and that a more professional editorial process is necessary before a final version can be accepted.

Response#8. We accept the comment and the question. And we have tried to edit as much as possible with online grammar and language checkers and with my friend fluent English speaker in my university.

 Thank you!!!

Fentahun Yenealem

---

## [Decision Letter · Decision Letter 4]

24 Jul 2020

PONE-D-19-27356R4

Intention to use vasectomy and its associated factors among married men in Debretabor Town, North West Ethiopia, 2019.

PLOS ONE

Dear Dr. Beyene,

Thank you for submitting your manuscript to PLOS ONE. After careful consideration, we feel that it has merit but does not fully meet PLOS ONE’s publication criteria as it currently stands. Therefore, we invite you to submit a revised version of the manuscript that addresses the points raised during the review process.

We look forward to receiving your revised manuscript.

Kind regards,

Wen-Jun Tu

Academic Editor

PLOS ONE

Reviewers' comments:

Reviewer's Responses to Questions

**Comments to the Author**

1. If the authors have adequately addressed your comments raised in a previous round of review and you feel that this manuscript is now acceptable for publication, you may indicate that here to bypass the “Comments to the Author” section, enter your conflict of interest statement in the “Confidential to Editor” section, and submit your "Accept" recommendation.

Reviewer #3: (No Response)

2. Is the manuscript technically sound, and do the data support the conclusions?

Reviewer #3: Yes

3. Has the statistical analysis been performed appropriately and rigorously? 

Reviewer #3: Yes

4. Have the authors made all data underlying the findings in their manuscript fully available?

Reviewer #3: Yes

5. Is the manuscript presented in an intelligible fashion and written in standard English?

Reviewer #3: Yes

6. Review Comments to the Author

Reviewer #3: The authors have addressed adequately the few substantive questions/comments I raised in the last round of review. I do not request any additional substantive edits. However, I still feel that the manuscript needs copyediting before it could be published. I do not know if PLoS ONE provides that service. If not, then it is worth noting that some formatting issues and grammatically awkward phrases remain in this revision. I don't believe this is the task of a reviewer to undertake all of these, as I illustrated previously with edits to the Abstract. I thus would ask that some formal copyediting process beyond what the authors have done be undertaken to ensure a final polished version of the manuscript.

7. PLOS authors have the option to publish the peer review history of their article (what does this mean?). If published, this will include your full peer review and any attached files.

Reviewer #3: No

---

## [Author Response · Author response to Decision Letter 4]

13 Aug 2020

To Plose One Academic Editor 

PONE-D-19-27356R4

From Fentahun Yenealem

First of all I would like to say thank you very much for your timely response and your actively involvement on the comments and coordination without hesitating by the current issue (COVID 19). And my heart full gratitude again reaches to reviewer#3 that was giving a measurable, fruitful and observable comments and questions by devoting their golden time and with the challenge of COVID19. 

I. Point by point response to Editor and Reviewer

Q# 1. Reviewer #3: The authors have addressed adequately the few substantive questions/comments I raised in the last round of review. I do not request any additional substantive edits. However, I still feel that the manuscript needs copyediting before it could be published. I do not know if PLoS ONE provides that service. If not, then it is worth noting that some formatting issues and grammatically awkward phrases remain in this revision. I don't believe this is the task of a reviewer to undertake all of these, as I illustrated previously with edits to the Abstract. I thus would ask that some formal copyediting process beyond what the authors have done be undertaken to ensure a final polished version of the manuscript.

Response 1: We accept the comment and the question. And we have tried to copy edit as much as possible with online grammar and language checkers and with my friend fluent English speaker in my university.

 Thank you!!!

Fentahun Yenealem

---

## [Editor Report · Decision Letter 5]

14 Aug 2020

Intention to use vasectomy and its associated factors among married men in Debretabor Town, North West Ethiopia, 2019.

PONE-D-19-27356R5

Dear Dr. Beyene,

We’re pleased to inform you that your manuscript has been judged scientifically suitable for publication and will be formally accepted for publication once it meets all outstanding technical requirements.

Kind regards,

Wen-Jun Tu

Academic Editor

PLOS ONE
---

## [Editor Report · Acceptance letter]

24 Aug 2020

PONE-D-19-27356R5 

Intention to use vasectomy and its associated factors among married men in Debretabor Town, North West Ethiopia, 2019. 

Dear Dr. Beyene:

I'm pleased to inform you that your manuscript has been deemed suitable for publication in PLOS ONE. Congratulations! Your manuscript is now with our production department. 

Kind regards, 

on behalf of

Dr. Wen-Jun Tu 

Academic Editor

PLOS ONE